# The Role of Cytokinins during the Development of Strawberry Flowers and Receptacles

**DOI:** 10.3390/plants12213672

**Published:** 2023-10-25

**Authors:** Moises Pérez-Rojas, David Díaz-Ramírez, Clara Inés Ortíz-Ramírez, Rosa M. Galaz-Ávalos, Víctor M. Loyola-Vargas, Cristina Ferrándiz, Ma. del Rosario Abraham-Juárez, Nayelli Marsch-Martínez

**Affiliations:** 1Departamento de Biotecnología y Bioquímica, Centro de Investigación y de Estudios Avanzados del Instituto Politécnico Nacional, Irapuato 36824, Mexico; moises.perez@cinvestav.mx (M.P.-R.); david.diaz@cinvestav.mx (D.D.-R.); 2Instituto de Biología Molecular y Celular de Plantas, Consejo Superior de Investigaciones Científicas—Universidad Politécnica de Valencia (CSIC-UPV), 46022 Valencia, Spain; clara.ortiz@ibmcp.upv.es (C.I.O.-R.); cferrandiz@ibmcp.upv.es (C.F.); 3Unidad de Bioquímica y Biología Molecular de Plantas, Centro de Investigación Científica de Yucatán, Mérida 97205, Mexico; gaar@cicy.mx (R.M.G.-Á.); vmloyola@cicy.mx (V.M.L.-V.); 4Departamento de Alimentos, División de Ciencias de la Vida, Universidad de Guanajuato, Irapuato 36500, Mexico; mabraham@ugto.mx

**Keywords:** cytokinin, strawberry flower and fruit, receptacle, development, Fragaria, benzyladenine (BA)

## Abstract

Cytokinins play a relevant role in flower and fruit development and plant yield. Strawberry fruits have a high commercial value, although what is known as the “fruit” is not a “true” botanical fruit because it develops from a non-reproductive organ (receptacle) on which the true botanical fruits (achenes) are found. Given cytokinins’ roles in botanical fruits, it is important to understand their participation in the development of a non-botanical or accessory “fruit”. Therefore, in this work, the role of cytokinin in strawberry flowers and fruits was investigated by identifying and exploring the expression of homologous genes for different families that participate in the pathway, through publicly available genomic and expression data analyses. Next, trans-zeatin content in developing flowers and receptacles was determined. A high concentration was observed in flower buds and at anthesis and decreased as the fruit approached maturity. Moreover, the spatio-temporal expression pattern of selected *CKX* genes was evaluated and detected in receptacles at pre-anthesis stages. The results point to an important role and effect of cytokinins in flower and receptacle development, which is valuable both from a biological point of view and to improve yield and the quality of this fruit.

## 1. Introduction

Fruits and seeds are one of the main sources of human food. In plants, fruits protect developing seeds and help disperse the new generation; these functions have allowed angiosperms to be the most dominant plant group in most ecosystems [1].

A botanical fruit results from fertilization of the ovary, part of the female reproductive organ, the gynoecium, and its final size is related to the number and size of the cells that form it. After pollination and ovule fertilization, new seed development and ovary growth as a fruit occur in synchrony [2,3]. The number of fruits a plant produces depends on the number of pollinated flowers. Flower number, development, and the number of cells in floral organs are mainly determined by meristematic activity. The number of cells, particularly in the ovary, affect final fruit size.

Hormones are some of the key factors that influence meristematic activity and other processes in plant development [4], including flower and fruit development, fruit ripening, and formation of viable seeds. Auxins, gibberellins, cytokinins, abscisic acid, and ethylene are involved in different stages of fruit development [5].

Cytokinins (CKs) play an important role in cell cycle control and participate in organ development programs. They can promote cell division, as first discovered in tissue culture assays, and are also involved in shoot activation, embryo development, stimulating leaf expansion due to increased cell size, and preventing leaf senescence [6,7].

Many functions were discovered by studying the effect of increased or decreased activity of cytokinin-degrading enzymes (cytokinin oxidases/dehydrogenases, CKXs) [8]. One of these functions is the regulation of fruit development [9,10,11,12,13].

There are several important connections and interactions between CKs and master regulators of meristematic activity. WUSCHEL (WUS), which is a key transcription factor in the control of meristematic activity, represses genes of the *ARABIDOPSIS RESPONSE REGULATOR* (RR) type A family, which attenuates the response to CKs [14], whereas SHOOTMERISTEMLESS (STM), an essential factor for meristem maintenance, promotes the expression of the *isopentenyltransferase* 7 (*IPT7*) gene, which encodes for an enzyme involved in CKs biosynthesis [15,16]. It has been reported that CKs response factors, ARR-Bs, activate *WUS* in the presence of this hormone [17]. The content of CKs in the meristem affects meristem activity, the number of flowers, and the size of floral organs [9].

Cytokinins are adenine derivatives with an isoprenoid chain at N^6^, although there are also cytokinins that contain an aromatic ring instead of the isoprenoid chain. Generally, for the isoprenoid types, CK biosynthesis starts with the transference of an isoprenoid chain from dimethyl-allyl-diphosphate (DMAPP) to the adenine bases in adenosine nucleotides (mostly ATP or ADP). Isopentenyltransferase enzymes catalyze this reaction. CYP735A enzymes then convert the resulting molecule to ribosides and ribotides. Afterwards, the action of the LONELY GUY (LOG) enzymes converts the ribosides into free and active forms of the hormone trans-zeatin [18]. Cytokinin signaling occurs primarily as a phosphorylation cascade. It begins with the hormone binding to the CHASE domain of Histidine kinase (HK) receptors found in the endoplasmic reticulum or in the plasma membrane. This triggers HK autophosphorylation, followed by the transfer of the phosphate group to Histidine phosphotransferases (AHPs). AHPs, in turn, transfer the phosphate group to cytokinin Response Regulators (RRs). Among the AHPs, pseudohistidine phosphotransferases (PHPs) have amino acid changes that impair their ability to transfer the phosphate, converting them into negative regulators of CK signaling. Type B RRs, once activated by phosphorylation, act as transcriptional regulators. They promote the transcription of Type A *RR* genes. Type A RRs can also receive the phosphate group from AHPs but lack a DNA-binding domain, so they act as negative regulators of the signaling cascade [18,19,20].

Cytokinin dehydrogenases (CKXs) catalyze the degradation of CKs, causing their irreversible inactivation. Reversible inactivation occurs through the conjugation of sugars and/or amino acids in the isoprenoid chain, where UGT enzymes (glycosyl transferase) participate. However, when these molecules are conjugated in the CK ring, their inactivation is irreversible. CKs are transported from the roots to the shoots through the xylem and from the shoots to the roots through the phloem, and, in the cells, they are transported through ABC, purine permease (PUP), or equilibrative nucleoside (ENT) transporters [18,19,20,21,22].

As CKX catalyzes the degradation and inactivation of CKs, they control local CK levels and contribute to the regulation of CK-dependent processes [23]. In *Arabidopsis thaliana*, by studying a *ckx3 ckx5* double mutant, it was found that higher CK content affected reproductive meristems, increasing the number of flowers and organ size and the number of seeds per fruit [9]. Similar results were reported in *Brassica napus*, as larger inflorescence meristems and increased seed production were observed in *ckx* mutants or when CKs were applied [10,24]. In rice, barley, and cotton, seed yield was also increased, relating to increased CKs in inflorescence meristems due to silenced or reduced expression of *CKX* genes [25,26,27]. Therefore, it is clear that CKs play a significant role in flower and fruit number and size.

Strawberry (*Fragaria × ananassa*) is a highly appreciated and cultivated “fruit” in many countries. While a typical botanical fruit develops from the gynoecium, in the case of the strawberry, what we know as “fruit” is an “accessory fruit” because it does not develop from a reproductive organ. The tip of the stem on which the floral organs are located, called the receptacle, enlarges and ripens as a fruit after fertilization of the ovaries. It becomes the commercial fruit, and the actual botanical fruits (achenes) are located on the surface of the receptacle. Each achene is a fertilized ovary containing a seed within the testa [28,29].

The genus Fragaria contains 21 known species. Throughout evolution, several of the diploid species have undergone a process of polyploidization. However, due to human intervention, other genotypes have been generated. One of them is *F. × ananassa*, the commercial strawberry, which was obtained by crossing two octaploid species: *F. chiloensis* and *F. virginiana* [30]. Strawberry is a member of the Rosaceae family, which includes other species of economic interest such as apple (*Malus domestica*), cherry (*Prunus avium*), almond (*Prunus dulcis*), rose (*Rosa* spp.), and raspberry (*Rubus ideaus*), among others. Unlike other family members, the strawberry grows and produces fruit faster and can be propagated vegetatively. These characteristics place strawberry as a suitable study model for Rosaceae [30].

Wild strawberry *F. vesca* is a diploid species (2n = 2x = 14). It has been reported to be easily transformable, adaptable to diverse climatic conditions, has a shorter life cycle than *F. × ananassa*, and is a valuable gene discovery and functional validation model. In addition, the sequenced genome (240 Mb) is available and contains 34,809 reported genes, so it is used as a study model for *F. × ananassa* and other species of the same family [29]. Though *F. vesca* fruits differ from commercial strawberry fruits in size and other characteristics, the availability of transcriptomic information from different stages and tissues is precious to identify candidate biological processes and genes that can be involved in fruit development [29].

Given the important role of CKs in the development of botanical fruits, in this work, we investigated the role of cytokinins during the development of strawberry accessory fruits and flowers. We identified homologous of CK genes that code for enzymes that participate in cytokinin metabolism and genes that code for elements of the two-component system, the cytokinin signaling pathway, in the genome of *F. vesca*. We explored their expression during fruit development using publicly available transcriptomic data. Moreover, we determined CK content in different flower and fruit developmental stages. Using the obtained information as a guide, we tested different cytokinin treatments and studied in detail the expression of selected *CKXs* during receptacle development in commercial strawberries.

## 2. Results

### 2.1. Cytokinin Genes Are Expressed during Flower and Receptacle Development

To explore the role of the CK pathway during flower and fruit development in strawberry, we first investigated in silico the expression of cytokinin genes using a publicly available *F. vesca* expression database [29]. For this, we searched *F. vesca* homologous genes using *Arabidopsis* loci identifiers in the database of plant genomes Plaza [31] https://bioinformatics.psb.ugent.be/plaza (accessed on 20 March 2021).

Table 1 contains the identifiers of *F. vesca* genes homologous to *Arabidopsis* genes belonging to the most important gene families participating in the CK pathway, including metabolism and signaling. For the CK biosynthesis pathway, there are seven genes homologous to *Isopentenyltransferase* (*IPT*) and nine genes homologous to *LONELY GUY* (*LOG*) genes (Table 1). From the signaling pathway, we found four genes homologous to *Histidine kinase* (*HK*), nine genes homologous to *Histidine phosphotransferase* (*HP*), seven genes homologous to *Response regulator type A* (*RR* type *A*), seven genes homologous to the *Response regulator type B* (*RR* type *B*), and four genes homologous to *Response regulator type C* (*RR* type *C*) genes. Regarding CK degradation, eight genes homologous to *Cytokinin dehydrogenase* (*CKX*) genes were found. The genes we found coincided with most of the previously reported genes for the *IPT*, *LOG*, *RR*, and *CKX* gene families [32,33,34]. More homologous genes were found, for example, for the *CKX* family. However, some of these homologous genes code for truncated proteins and were not included in the following analyses.

The model plant *Arabidopsis*, where the cytokinin signaling pathway has been mostly studied, has three histidine kinases that act as cytokinin sensors, AtHK2, 3, and 4, with AtHK4 performing a predominant role. The CHASE domain is responsible for this sensing capacity [35]. A related kinase, AtHK1, is proposed to act as an osmosensor [36,37]. We found related HKs in the genome of *F. vesca*, though only FvHK2, 3, and 4 appeared to be the closest in sequence of the CHASE domain of Arabidopsis cytokinin receptors AHK2, 3, and 4. In contrast, FvHK1 appeared closer to AHK1 (Figure 1A,C).

In *Arabidopsis*, six genes of the AHP family are known to be involved in the signaling pathway; however, in *F. vesca*, we found nine homologous genes. The phylogenetic tree of *Arabidopsis* and *F. vesca* HP proteins shows that AtAHP6 and FvHP6 are very close (Figure 1B). AtAHP6 lacks a highly conserved histidine residue (His, H), present in all other HPs. Instead, it is replaced by an asparagine (Asn, N), whereas the H residue is found two positions earlier. Due to these changes, the phosphorylation cascade is interrupted, and the transcriptional response to CK does not occur or is reduced. Therefore, AHP6 is considered to act as a negative regulator. When we compared the amino acid sequence, FvHP6 was very similar, with the Histidine two positions earlier and the asparagine residue in the position of conserved H (Figure 1D). In the case of FvHP7, it also lacks the conserved H residue, although instead of N, it is replaced by an aspartic acid (Asp, D), and leucine (Leu, L) was observed two positions before instead of the histidine present in AHP6 and FvHP6 (Figure 1D).

We then used the *F. vesca* identifiers to obtain publicly available RNA-seq data for each gene [29]. All the studied genes of the CK pathway were expressed at different developmental stages in diverse vegetative and reproductive tissues (Appendix A). We next focused on those tissues that are related to strawberry flowers and fruits, in this case, carpels, pith, flower, and early stages of receptacle development (Figure 2). Many homologous genes to those that participate in the CK pathway (biosynthesis, signaling, and degradation) were found to be expressed in these tissues. For example, the CK biosynthesis genes *FvIPT1* and *2* and *FvLOG2* and *3*; the cytokinin signaling genes coding for receptors *FvHK2*, *3*, and *4*; *FvHP4* and 6, and type B1 and type A3 *RRs*; and cytokinin-inactivating genes such as *CKX7* are expressed in the flower and early stages of receptacle development. At later developmental stages, after anthesis, the number of expressed cytokinin genes was less than at earlier stages, but, still, some genes, such as *FvHKs*, *FvRRs*, *FvLOGs*, and *CKXs*, were expressed.

This indicates that the cytokinin pathway may participate in the development of strawberry flowers and that it is active and most likely participates in the receptacle development.

### 2.2. Cytokinin Content Diminishes after Anthesis

After observing the gene expression changes, we sought to investigate the changes in cytokinin content during flower and fruit development in *F. × ananassa.* For this, the trans-zeatin content was measured to determine the hormone concentrations at different stages of development, from flower buds to ripe fruits obtained from strawberry plants grown in the field. The analyzed samples were flower buds, flowers at anthesis, and receptacles 8, 13, and 21 days after anthesis.

Flower buds contained an average of 5416.3 pmol of trans-zeatin g DW^−1^. In contrast, flowers at anthesis contained 4653.25 pmol g DW^−1^, receptacles 8 days after anthesis 287.6 pmol g DW^−1^, and receptacles 13 days after anthesis 160.5 pmol g DW^−1^ (Figure 3). The CK levels in receptacles 21 days after anthesis (stage very close to harvest) were very low or null (not included in the graph). From these analyses, it was clear that, in our conditions, hormone content was higher at the early stages of development and decreased as the fruit (receptacle) approached harvest. In this experiment, CK concentration is up to 30 times higher in flower buds at early stages than in advanced stages of fruit development.

### 2.3. Exogenous Application of Cytokinins in Strawberry Plants Impact Flower Number and Fruit Size and Weight

Cytokinins have been applied to different crops to increase yield and other important agronomic characteristics in botanical fruits, so exogenous applications of this hormone to strawberry plants was carried out to investigate the effect in field conditions (same conditions as used for cytokinin determinations in flowers and developing fruits). Cytokinin applications were tested at two different periods: fall–winter and spring–summer (Table 2). In both cases, flowers were marked from the time of anthesis until harvest to follow fruit development. During the fall–winter, the time from anthesis to harvest was 32–35 days (Figure 4).

The treatments included repeated 50 ppm BA applications with intervals of one week (one to four applications, T2 to T5 and T7), with intervals of two weeks (two applications, T6), or 100 ppm BA. Sixty-five days after the first application (DAFA), the plants treated four times with 50 ppm BA (T5) had a higher number of flowers (Figure 5E) than control plants. On the other hand, those treated with three exogenous applications (T4) of the hormone (50 ppm) at intervals of 7 days between applications presented the highest values in fruit width and weight when compared to the control and the other applications (Figure 5A–C). The highest total number of fruits was obtained when applying 50 ppm once (Appendix A).

Next, the application experiment was performed at a different period of the year, spring–summer. At this time of the year, when there was a natural increase in temperature and hours of light during the day, in our conditions, the time lapse from anthesis to harvest was shorter compared to the fall–winter experiment, with an average of 21–23 days. Moreover, an increase in the total number of flowers was observed compared to the number of flowers formed in the fall–winter experiment, regardless of the treatment. Almost twice the number of flowers per plant was produced in the spring–summer compared to the fall–winter experiment. 

This experiment evaluated three treatments: (a) 50 ppm BA applications one and three times at seven-day intervals because these treatments produced interesting results in the winter experiment, and one 100 ppm application to compare the effects of a higher concentration. In total, 35 days after the first application, the fruits of plants that had received the different treatments had increased length were wider or heavier depending on the treatment than the control fruits (Figure 6A–C).

At 63 DAFA, a clear increase in the number of flowers was observed in all plants where cytokinins were applied, compared to the control, and this was more evident for the plants that were treated once with 50 ppm BA (Figure 6D,E).

Moreover, at 63 DAFA, the fruits of plants sprayed with CKs presented greater dimensions (length and width) and weight when compared to the fruits of control plants (Figure 6A–C). All CK treatments resulted in longer, wider, and heavier fruits than control fruits in these conditions. Also, the total weight of all the fruits produced in each of these treatments was higher than the control (Appendix A).

From the above experiments, we concluded that depending on the treatment and time of evaluation, exogenous application of BA can increase the number of flowers on strawberry plants and influence fruit dimensions and weight. Moreover, the change in climatic conditions alone affected the number of strawberry flowers and fruits developed and the sizes and weights of fruits.

### 2.4. Spatio-Temporal Gene Expression Analysis of CKX Genes in Developing Receptacles of F. × ananassa

The CK exogenous treatments showed interesting effects on strawberry development, and it would be interesting to test the effect of increasing endogenous CK levels. A strategy to increase these levels would be to create loss-of-function mutations of the CK-degrading enzymes CKXs. As a first step for future experiments, we sought to visualize, using the in situ hybridization technique, the expression of *CKXs* genes during the development of the strawberry receptacle to find those that could be interesting targets for gene editing. *IPT2* and *WUS1* were also included in these analyses. Based on the expression pattern observed in *F. vesca*, orthologs for three candidate *CKX* genes that participate in the CK degradation pathway were selected: *CKX2*, *CKX3*, and *CKX7*. *IPT2* was selected as a CK biosynthesis gene. In the case of *WUS*, its expression had been reported in *F. vesca*, and the intention was to compare its expression in *F. × ananassa*.

The Genome Database Rosaceae (GDR) page was used to obtain the sequences of the *F. × ananassa* orthologous genes. As *F. × ananassa* is an octaploid plant, several copies of similar genes were found, and only those with the greatest similarity were chosen (Table 3). 

Flower buds at developmental stages 9–11 were used to study the expression of the genes. Interestingly, expression for both *CKX2* and *CKX7* was observed in flowers at stages 10 to 11. At stage 10, the standard bottle-shaped carpel morphology was observed, and the meiosis of MMC (Megaspore mother cells) is probably occurring as reported for *F. vesca*. In stage 11, rapid elongation of the style, accompanied by ovary expansion, occurs, and carpels shaped like a music note can be observed at the receptacle base. The carpels reach their maximum sizes, the ovary is completely formed, and the stigmas become scalloped [38].

In the case of *CKX2*, transcripts were detected in the receptacle at stages prior to anthesis, as seen in Figure 7A–C. The expression appears in the central pith and the base of the receptacle at stages 10 and 11, becomes mainly confined to the central pith, and expands slightly to the cortex in stage 11. *CKX7* transcripts were also detected in the central pith, though at a lower accumulation, and were more confined to the pith center and base (Figure 7D–F). Expression of neither *CKX2* nor *CKX7* was detected at earlier stages.

In the case of *CKX3*, *IPT2*, and *WUS*, the probe signal was not detected at the different stages evaluated (Figure 8A–C).

## 3. Discussion

### 3.1. Cytokinin Related Gene Expression and Cytokinin Content Change during Flower and Receptacle Development

In this work, we explored the role of CKs in strawberry receptacles and general flower development. We started by exploring genes of the cytokinin pathway in the data available for *F. vesca* [29,39]. Cultivated strawberry (*F. × ananassa*) arose from the hybridization of the genomes of *F. virginiana* and *F. chiloensis*, two wild octoploid species. These species were formed by the fusion and interactions of the genomes of four wild diploid progenitors. From these original wild progenitor genomes, the *F. vesca* subgenome is the dominant one, meaning that that it contributes with the most highly expressed homoeologous (genes duplicated after the polyploidization) that the other three genomes [40].

The *F. vesca* genome contains similar numbers of genes for biosynthesis genes (*IPTs* and *LOGs*, [32], *ARRs* type A, B, and C [33], and genes coding for enzymes responsible for degradation (CKX, [34]) as the genome of the model plant *A. thaliana*. For the cytokinin receptors (Histidine kinases, HKs), we found four genes in the *F. vesca* genome close to the *Arabidopsis* AHKs related to cytokinin. From these receptors, FvHK2, 3, and 4 may be acting as true CK receptors since their CHASE domain, the domain responsible for sensing CKs [35], is similar to the functional receptors in *A. thaliana* AtHK2, 3, and 4. From the other FvHKs considered, FvHK1 has a domain similar to AtHK1, so it may act as an osmosensor [36,37].

Regarding the elements responsible for the next step of cytokinin signaling, Histidine Phosphotransferases (HPs), nine genes were found in the *F. vesca* genome, whereas the *A. thaliana* genome contains six. AtHPs receive the phosphate from the HKs and transfer it to the RRs, except for AtHP6, which cannot transfer it [19,41]. FvHP6 may function as an equivalent to the negative regulator AtHP6 in Arabidopsis based on the similarity of the changes in the H position in the domain concerning other HPs. Interestingly, a different HP, FvHP7, lacks a key Histidine in the conserved domain. Therefore, it may be functioning in a different way than the other HPs or HP6s.

When the expression of the genes belonging to different families was analyzed, we noticed a higher number of CKs genes showing expression at the flower and early receptacle stage than at later stages. Interestingly, some family members of the *IPT*, *LOG*, and ARR A families appear to be mainly expressed at earlier stages, before anthesis (such as *FvIPT1* and *2*), and some mostly at later stages, though at lower levels (like *FvIPT7*). In the case of the CK receptors, *FvHK2* and *3* are expressed early, whereas *FvHK1*, which may not act as a true CK receptor, is active at the later stages. For other families, like type B ARRs, most expression occurs only in the earlier stages. Future studies can cover other genes such as *UGTs* (Glycosyl transferases) [21,42], *KMDs* (Kiss me deadlys) [43], *CRFs* (Cytokinin response factors) [44] and others that also participate in the cytokinin pathway.

### 3.2. Cytokinin Concentration

The reduction trend in the expression in many gene families related to the cytokinin pathway in flowers and developing receptacles of *F. vesca* correlates with the trend observed in CK content in flowers and developing receptacles of *F. × ananassa*. In further studies, it will be very interesting to separate the different parts of the flower or the developing receptacle to clarify CK distribution and compare it to other studies [45]. Nevertheless, both the expression of genes that participate in the cytokinin pathway and the cytokinin content determinations suggest that the cytokinin pathway may be most relevant during the formation of flowers and receptacles of *F. vesca* and *F. × ananassa*, respectively, and that it is attenuated at later stages. Interestingly, though they do not include samples of flower buds and flowers at anthesis, there are studies that have also found that the trans-zeatin content is very low and even decreases during fruit development and ripening increased during fruit development [46], whereas others found that zeatin, in general, increases at the ripening stages [47]. It would be interesting to evaluate whether the increment found in the latter work is due to cis-zeatin.

### 3.3. Exogenous Application in the Field Led to Changes in Flower Number and Fruit Size and Weight

In our experiments carried out in unprotected strawberry orchards, the exogenous application of CKs caused an increase in the number of flowers and fruit size or weight, in treated plants, and this coincides with reports of other studies performed under protected conditions [45,48,49,50,51,52,53,54]. Depending on the number of applications and concentration, CK applications have an evident influence on fruit formation, size, and weight, which could translate into an increase in yield. This also resembles the increase in flower numbers and fruit size in *Arabidopsis* caused by an increased accumulation of endogenous CKs due to reduced CK degradation in loss-of-function mutants of genes of the *CKX* family [9]. The same effect, an increase in yield, has been reported in rice, cotton, and barley *ckx* mutants [25,26,27].

However, it appears that exogenous application of CKs does not have an immediate effect, i.e., applying CKs to fruits that are already developing does not affect their size. Instead, statistically significant differences regarding fruit characteristics and flower number were observed after more than four weeks, depending on the experiment. Therefore, it is very likely that CKs mostly affect early stages of flower development, impacting the development of the receptacle resulting in larger “fruits”. They also act even before flowers start developing since CK applications stimulated the formation of larger number of flowers.

Interestingly, in the case of our experiments, 32–35 days elapsed from anthesis to harvest in winter and 21–23 days in spring–summer. There was almost a 10-day difference from anthesis to a fruit ready to harvest. Also, almost twice the number of flowers was produced in spring–summer compared to the fall–winter period. This increase in flower number was also reflected in an increase in fruit number. However, in the spring–summer experiment, more fruits were harvested, but they were generally smaller and lighter than those harvested during the winter. It would be interesting to test the role of endogenous CKs in these large seasonal differences in future experiments since light affects cytokinin signaling and, therefore, meristematic activity [55].

Nevertheless, applied CKs clearly affect flower number and fruit size and weight in both seasons. Interestingly, not all treatments were equally effective. Most treatments led to higher yields, but not all. For example, in the fall–winter season, some treatments even led to smaller fruit sizes. This may be related to the different effects that CKs can have depending on the time of development of an organ, such as leaves or botanical fruits [56,57].

### 3.4. Spatiotemporal Expression Analysis of Cytokinin Pathway Genes in Developing Receptacles of F. × ananassa

The in silico expression pattern of genes involved in cytokinin biosynthesis, signaling, and degradation was used as a basis to select genes for the in situ hybridization expression analyses in *F. × ananassa* flowers and receptacles. In this species, there are several copies of the same gene, and probes were designed to detect the different copies. High *CKX2* and *CKX7* expression was observed in strawberry receptacles, which coincided with *F. vesca* transcriptome data, which further supported that the *F. vesca* data were useful to guide experiments in octaploid strawberries [29]. The expression of *CKX2* and *CKX7* occurred in the advanced stages of flower development before anthesis, and their expression was not observed in the early stages.

Expression of *CKX3* and *IPT2* was not detected in our assays. In the case of *CKX3*, we expected to find a similar pattern to *CKX2* and *CKX7*; however, it is possible that this gene was not expressed in that tissue at that time or that the probes used were not optimal. This could also be the case for *IPT2*.

For the evaluation of *WUS*, the same oligos reported by Li et al., 2019 [58] for *F. vesca* were used to make the probe, and, although a BLAST search was performed with the *F. × ananassa* sequences to confirm homology, the probe signal was not detected in the tissues evaluated in commercial strawberry. The lack of signal detection could be due to earlier expression (stages 3–8 of flower development, which were not included in this study).

Interestingly, from the equivalent *FvCKX*s for which expression was detected in *F. × ananassa* (marked in blue in Appendix A), we expected them to be close to the CKXs that produce higher yields when mutated in *Arabidopsis* and rice (marked in red in Appendix A). However, they belong to different clades. This difference might reflect that different genes perform equivalent roles in the receptacle as a non-reproductive tissue, compared to the reproductive tissues of rice and *Arabidopsis*.

## 4. Materials and Methods

### 4.1. Biological Material

CK quantification, and exogenous applications of CKs, were performed on strawberry plants growing in an open field. Commercial strawberry var. Camino Real was transplanted between July and August 2020 in a property in the ejido Lo de Juarez in Irapuato, Guanajuato, with coordinates 20°47′31.7” N 101°21′19.6” W.

For in situ *hybridization* (ISH) experiments, samples of greenhouse grown, pre-anthesis strawberry var. San Andreas flowers at different stages of development were used.

### 4.2. Gene Identification and In Silico Analyses

*A. thaliana* locus identifiers were used search for genes involved in the cytokinin pathway in *F. vesca* in Plaza: Comparative Genomics in Plants version 5.0 [31]. During this work, *F. vesca* genes from specific families (*ARRs* type A and B, *CKX*, *IPT*, and *LOG*) were reported [32,33,34], and they coincided with the ones we found. For the expression analyses, we only considered genes that were translated into complete proteins, leaving truncated proteins outside of this study. In Plaza, we obtained the identifiers (id), the nucleotide, and the amino acid sequence for each component of the cytokinin pathway. Phylogenetic analyses were performed for AHK, AHP, and CKX proteins using MEGA11 by Maximu Likelihood (MLE) method with 100 bootstraps [59]. The identifiers obtained in Plaza for each cytokinin gene were used to search for expression data in the publicly available *F. vesca* RNA-seq database [29]. This database contains transcriptome data from different tissues and developmental stages. The Morpheus^®^ software (https://software.broadinstitute.org/morpheus accessed on 12 March 2022) was used to make heat maps with the expression data.

### 4.3. Cytokinin Determination

CK content was determined in untreated, field-grown (Irapuato, Gto., Mexico) strawberry plants of var. Camino Real in the summer of 2021. Whole-flower buds from approximately seven days before anthesis, whole flowers at anthesis, and receptacles (including achenes) 8 and 13 days after anthesis were collected. Three independent biological replicates were assayed for each analysis.

CKs were determined using a method modified from Dobrev et al., 2017 [60]. For CK extraction, a 10 mg dry weight (DW) tissue sample was weighted and homogenized with liquid nitrogen in a mortar until a fine powder was obtained, then 1 mg of butylated hydroxytoluene (Acros Organics, 112992500, Thermo Fisher Scientific, Waltham, MA, USA), dissolved in one mL of ethyl acetate (CTR Scientific 00184), was added. Then, 0.5 mL cold (−20 °C) extraction solvent (methanol–water–formic acid = 15/4/1, *v*/*v*/*v*) was added, homogenized, collected directly into a microcentrifuge tube, and centrifuged (Hettich ZENTRIFUGEN, Mikro 22 R, Buford, GA, USA) at 20,000× *g* at 4 °C for 20 min. The supernatant was transferred into a new 2 mL tube, and the pellet was re-extracted with an additional 0.5 mL extraction solvent for 30 min and centrifuged as above. The pooled supernatants were evaporated in a SpeedVac (CentriVap microIR, LABCONCO) at 10 mBar and 40 °C until the sample was reduced to ¼ of the initial volume (less than 0.25 mL). Samples were frozen and stored at 80 °C until their purification.

Citokynin Purification. An Oasis MCX (1 mL/30 mg retention capacity; Waters 186000252, Wexford, Ireland) column was equilibrated with 1 mL methanol, followed by 1 mL solid-phase extraction (SPE) load solvent (1 M formic acid). The sample was diluted in 0.5 mL of SPE load solvent and applied to the column. The flow-through was discarded, the column was washed with 0.5 mL SPE load solvent, followed by 1 mL water, the flow-through was discarded, and 0.5 mL methanol (solvent 1) was applied. The flow-through was collected into a new 2 mL microcentrifuge tube. Next, 0.5 mL of solvent 2 (0.35 M ammonium hydroxide in 70% methanol) was applied, and the flow-through was collected into another new 2 mL microcentrifuge tube. This was the CK fraction. The fractions collected were evaporated in SpeedVac (CentriVap microIR, LABCONCO, Kansas City, MO, USA) at 10 mBar and 40 °C to dryness. Dried fractions were stored at −20 °C until LC-MS analysis.

HPLC Quantitative Analysis. The CK fraction was suspended in one mL of the HPLC mobile phase system for CK 30% acetonitrile HPLC (TEDIA UN1648) and 70% water HPLC (JT Baker 4218–03, Phillipsburg, NJ, USA) containing 0.5% acetic acid (CTR Scientific 00500, Mexico City, Mexico), filtered through a Millipore filter (0.22 µm). Chromatographic separation of CK was achieved on a Kromasil C_18_ reverse-phase column (5 µ, 100 Å, 250 × 4.6 mm, Phenomenex, Torrance, CA, USA). The detection was carried out at 280 nm with a diode array detector-3000RS as a part of an Ultimate 3000 UHPLC series (Thermo Scientific Dionex, Waltham, MA, USA) pump and a WPS-3000 autosampler. The column was operated at room temperature (25 °C). A flow rate of 1.0 mL min^−1^ was applied with an isocratic system. The injection volume was 20 μL. Chromeleon Chromatography Data System software v7 (Thermo Scientific, Waltham, MA, USA) was used to control the instrument.

### 4.4. Exogenous Application of Cytokinins in Strawberry Plants

The effect of the exogenous application of cytokinins was evaluated in commercial strawberry (*F. × ananassa*) var. Camino Real grown in an outdoor field. Seven treatment plans with applications of BA (6-benzyladenine) at doses of 50 parts per million (ppm) or 100 ppm were evaluated. Each treatment plan included three replicates of ten strawberry plants per replicate. Different application frequencies were evaluated (one, two, three, and four applications) with seven days intervals, except for T6, with a 14 day interval between applications. They were compared to control plants in which no cytokinins were applied. The experiment was repeated twice (Table 2).

The first experiment started on 2 December 2020 (fall-winter experiment). In each evaluation, flowers at anthesis were marked, the number of flowers at anthesis was counted on each evaluation date, the fruits were harvested, and the weights, lengths, and widths of each fruit were recorded. The evaluations were carried out at 0, 3, 7, 10, 14, 17, 21, 24, 28, 31, 35, 38, 42, 45, 49, 52, 55, 58, 62, and 65 days after the first application (DAFA), and they coincided with strawberry harvest days in the field. The following temperatures were recorded for December (Maximum: 31 °C; Minimum: 2.4 °C; Average: 17.7 °C) and January (Maximum: 31.2 °C; Minimum: −0.2 °C; Average: 16.9 °C).

The experiment was repeated in the same plot and with the same variety. Four treatments with four replicates each were evaluated with applications of BA at doses of 50 and 100 ppm. Each replicate consisted of ten strawberry plants. Different application frequencies (one and three applications with seven days intervals) were evaluated. A control without cytokinin application was also included. This experiment started on 26 April 2021 (spring–summer experiment). The evaluations were performed at 2, 4, 7, 9, 11, 14, 16, 18, 21, 23, 25, 28, 30, 32, 35, 37, 40, 42, 44, 46, 49, 51, 53, 56, 58, 61, and 63 days (DAFA) and coincided with the harvest of strawberries. The following temperatures were recorded May (Maximum: 35.3 °C; Minimum: 8.1 °C; Average: 23.1 °C) and June (Maximum: 34.4 °C; Minimum: 11.1 °C; Average: 21.6 °C).

#### 4.4.1. Statistical Analysis

Statistical analyses and graphs were performed with R-Studio (“RStudio: Integrated Development for R” v2022. Rstudio PBC, Boston, MA, USA, 2022), using Kruskall-Wallis tests for non-parametric variables and ANOVA for parametric variables. Post hoc tests were used when significant differences were found (Dunnett’s test for non-parametric variables and LSD for parametric variables).

#### 4.4.2. Spatio-Temporal Expression Analysis (In Situ Hybridization)

The in situ hybridization (ISH) protocol used was based on the work published by Ferrándiz et al., 2000 [61]. *CKX*, *WUS*, and *IPT* gene expression was analyzed in *F. × ananassa* fruits and flowers (San Andreas variety grown in greenhouse conditions in Valencia, Spain). The main criterion for specific gene selection for the hybridization experiments was the level and location of expression of homologous genes, as visualized with the *F. vesca* eFP Browser tool v.1.6.0 [29].

The GDR database (Genome Database for Rosaceae, www.rosaceae.org (accessed on 10 June 2022) [62] was used to obtain the sequences required to design specific oligos to clone fragments of the selected homologous *F. × ananassa* genes to make the probes (Table 3). *F. × ananassa* tissues were collected at stages 6–12 based on the morphological description of flower development [38]. Three tissue samples were used for each stage evaluated.

The Primer3Plus tool v3.3.0 was used to design the primers with the following characteristics: (1) the fragment to be amplified should be 200–500 bp, and (2) the fragment should be specific for each gene, but, at the same time, it should be able to amplify a fragment of the copies of the same homeologous genes. For example, the primers for *CKX2* would amplify a fragment for the eight octaploid strawberry copies but could not amplify the *CKX3* and *CKX7* genes. The T7 promoter sequence was added to the reverse primer.

## 5. Conclusions

The results of the gene expression data mining, CK content quantification, and CK application experiments in strawberry plants indicate that CKs are involved in developing strawberry flowers and receptacles, particularly before anthesis. *F. vesca* data helped guide expression analyses in *F. × ananassa*. Finally, the expression of *CKX2* and *CKX7* in developing receptacles makes them interesting candidates for gene editing.

## Figures and Tables

**Figure 1 plants-12-03672-f001:**
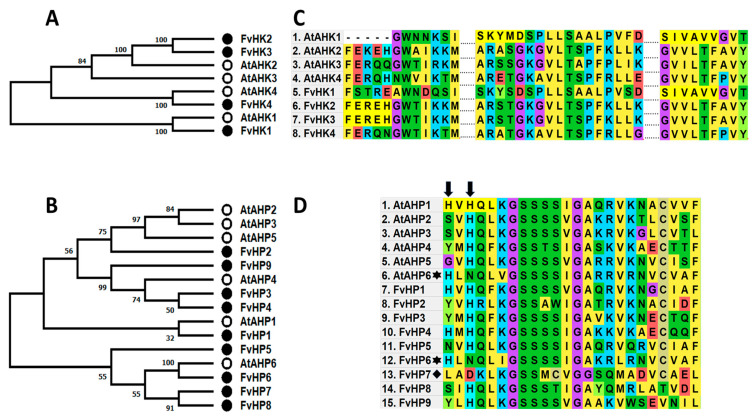
Phylogenetic trees and sequence alignments of conserved domains of HK and HP proteins by MLE method and 100 bootstraps. (**A**) Phylogenetic tree of HK protein sequences in *A. thaliana* and *F. vesca*. (**B**) Phylogenetic tree of *A. thaliana* and *F. vesca* HPs. (**C**) Amino acid alignment of a region of *A. thaliana* and *F. vesca* HK CHASE domains. (**D**) Amino acid alignment of *A. thaliana* and *F. vesca* HP sequences. White circles indicate *A. thaliana*, and black circles *F. vesca* sequences. Arrows indicate the conserved H and the displaced H in AtAHP6 and FvHP6, marked by asterisks. The diamond indicates FvHP7, where an aspartic acid (Asp, D) replaces the conserved H, and an L replaces the displaced H.

**Figure 2 plants-12-03672-f002:**
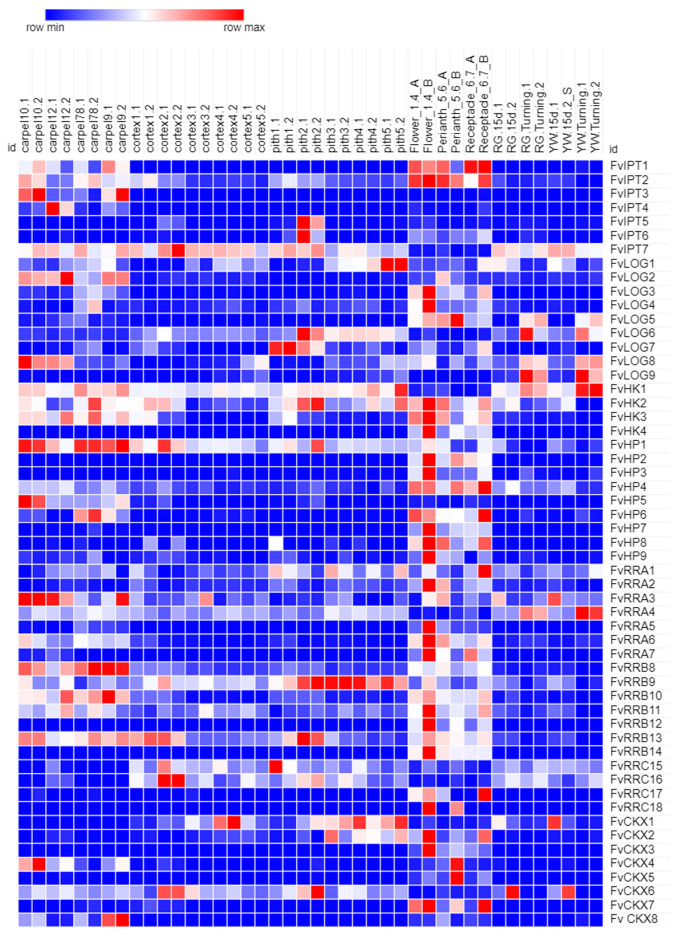
Heatmap-depicting transcriptome data of *F. vesca* floral organs and receptacles at different developmental stages. Transcriptomic data of Carpels, Cortex, Pith, Flower, Perianth, Young receptacle, Rugen (RG), and Yellow wonder (YW) receptacles at 15 days after anthesis (15d) and turning were obtained from Hawkins et al., 2017 [29].

**Figure 3 plants-12-03672-f003:**
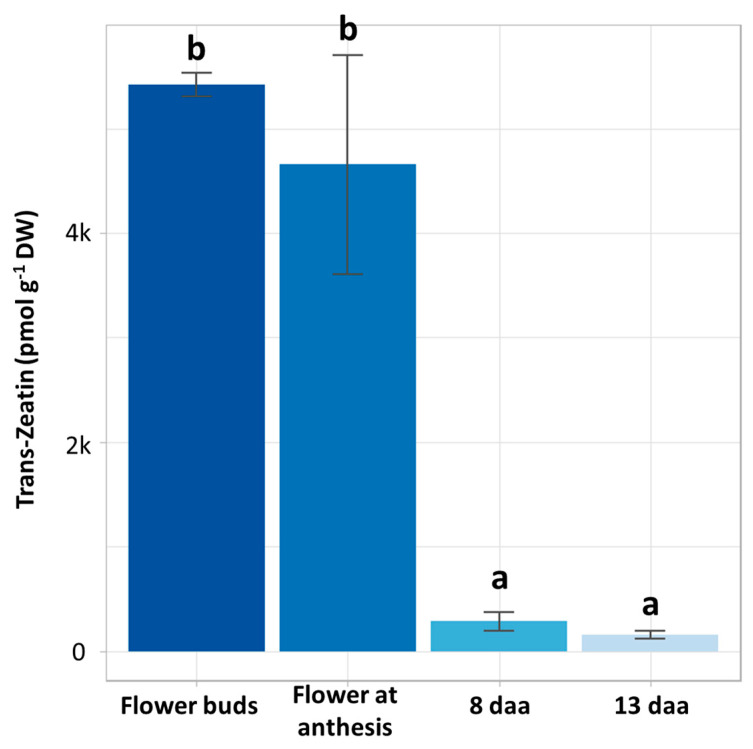
Determination of the concentration of trans-zeatin by HPLC-MS at different stages of flower and fruit development of strawberry (*F. × ananassa*). Flower buds, flowers at anthesis, and receptacles 8 and 13 days after anthesis (daa). pmol of trans-zeatin g dry weight^−1^. Different letters indicate statistically significant differences.

**Figure 4 plants-12-03672-f004:**
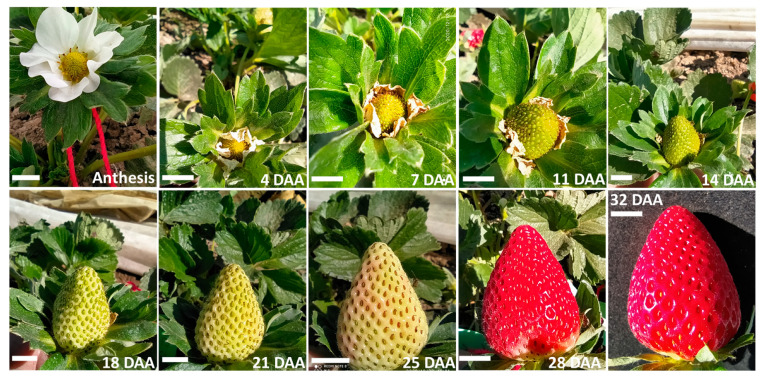
Strawberry development from anthesis to fruit harvest. The photographs depict control flowers and fruits evaluated during the exogenous cytokinin application experiment (fall–winter) from Camino Real variety plants grown in Irapuato, Guanajuato, Mexico. Development was followed from December 2020 to January 2021. Days after the anthesis (DAA) are indicated. Bar = 1 cm.

**Figure 5 plants-12-03672-f005:**
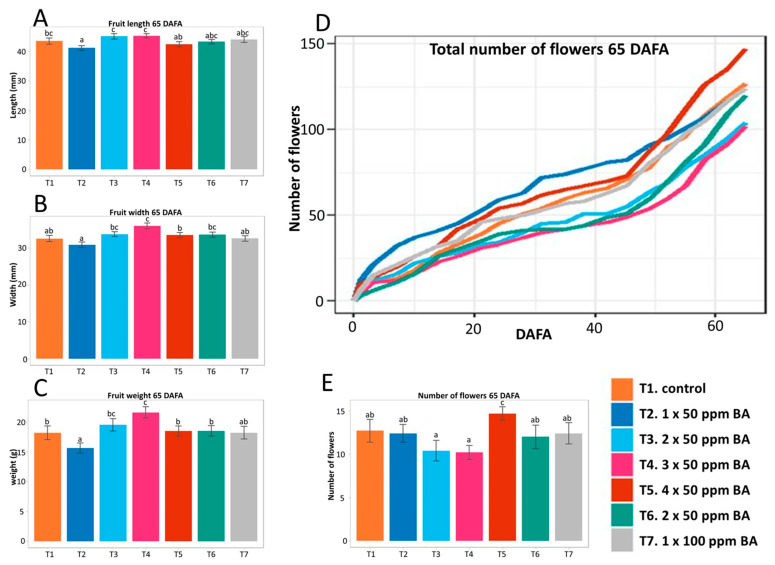
Exogenous application of BA in strawberry, fall-winter experiment. Evaluation and statistical analysis of length (**A**), width (**B**), and fruit weight (**C**) at 65 days after the first application (DAFA). Cumulative total number of flowers in each treatment (**D**) from 0 to 65 DAFA. Number of flowers formed per plant (**E**) at 65 DAFA. Different letters indicate statistically significant differences.

**Figure 6 plants-12-03672-f006:**
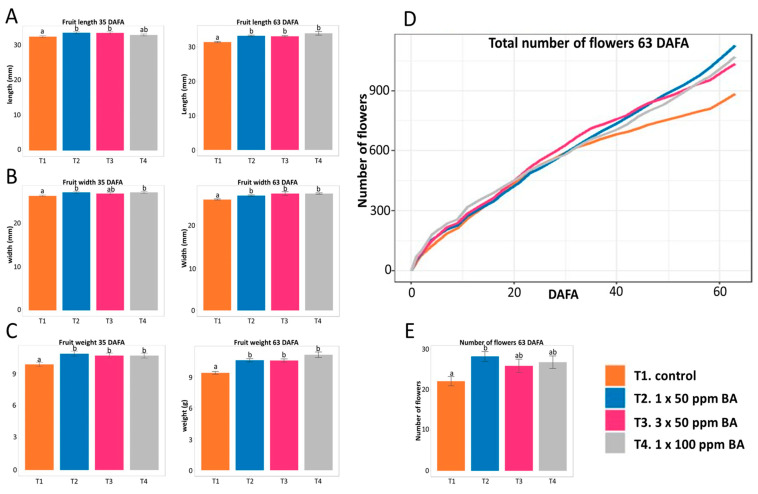
Exogenous application of BA in strawberry plants in the spring–summer experiment. Evaluation and statistical analysis of length (**A**), width (**B**), and fruit weight (**C**) at 35 and 63 days after the first application (DAFA). Accumulated flowers in each treatment (**D**) at 63 DAFA. Number of flowers formed per plant (**E**) at 63 DAFA. Different letters indicate statistically significant differences.

**Figure 7 plants-12-03672-f007:**
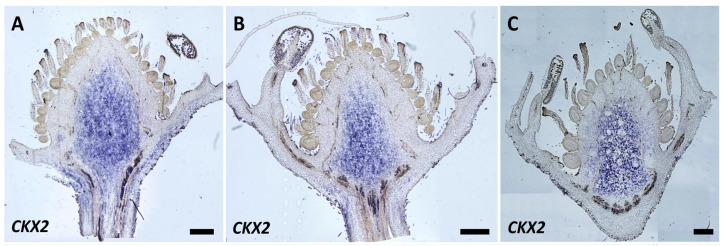
*CKX2* and *CKX7* in situ hybridization. Detection of the probe corresponding to the *CKX2* (**A**–**C**) and *CKX7* (**D**–**F**) genes stage 10–11 of flower development [38] of *F. × ananassa* var. San Andreas. Bar = 500 μm.

**Figure 8 plants-12-03672-f008:**
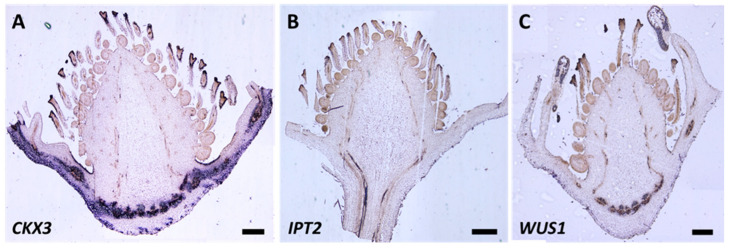
RNA in situ hybridization. Detection of the probe corresponding to the *CKX3* (**A**), *IPT2* (**B**), and *WUS* (**C**) genes in flowers of *F. × ananassa* var. San Andreas at developmental stage 10–11. Bar = 500 μm.

**Table 1 plants-12-03672-t001:** Homologous genes of the cytokinin pathway in *F. vesca*. Some of the *F. vesca* genes had been already reported and are included in this table.

Biosynthesis *				Signaling			
	Isopentenyl transferase		Lonely guy		Histidine Kinase		Histidine Phospho transferase
Identifier	*IPT*	Identifier	*LOG*	Identifier	*HK*	Identifier	*HP*
FVE03725	*FvIPT1*	FVE16128	*FvLOG1*	FVE14583	*FvHK1*	FVE06646	*FvHP1*
FVE23509	*FvIPT2*	FVE30641	*FvLOG2*	FVE21593	*FvHK2*	FVE27751	*FvHP2*
FVE30601	*FvIPT3*	FVE09744	*FvLOG3*	FVE30742	*FvHK3*	FVE09006	*FvHP3*
FVE27343	*FvIPT4*	FVE15306	*FvLOG4*	FVE04136	*FvHK4*	FVE25730	*FvHP4*
FVE27842	*FvIPT5*	FVE08500	*FvLOG5*			FVE22351	*FvHP5*
FVE06080	*FvIPT6*	FVE02714	*FvLOG6*			FVE15747	*FvHP6*
FVE27180	*FvIPT7*	FVE30448	*FvLOG7*			FVE25334	*FvHP7*
		FVE30673	*FvLOG8*			FVE30249	*FvHP8*
		FVE30477	*FvLOG9*			FVE11292	*FvHP9*
**Signaling ***						**Degradation ***	
	RR type A		RR type B		RR type C		Cytokinin Oxidase/Dehydrogenase
Identifier	*RRA*	Identifier	*RRB*	Identifier	*RRC*	Identifier	*CKX*
FVE00726	*FvRRA1*	FVE06245	*FvRRB8*	FVE08408	*FvRRC15*	FVE14452	*FvCKX1*
FVE09501	*FvRRA2*	FVE06943	*FvRRB9*	FVE08409	*FvRRC16*	FVE30654	*FvCKX2*
FVE11196	*FvRRA3*	FVE05178	*FvRRB10*	FVE16714	*FvRRC17*	FVE21961	*FvCKX3*
FVE14296	*FvRRA4*	FVE10041	*FvRRB11*	FVE08468	*FvRRC18*	FVE12649	*FvCKX4*
FVE21358	*FvRRA5*	FVE13325	*FvRRB12*			FVE12648	*FvCKX5*
FVE21885	*FvRRA6*	FVE15029	*FvRRB13*			FVE15382	*FvCKX6*
FVE28096	*FvRRA7*	FVE15958	*FvRRB14*			FVE30442	*FvCKX7*
						FVE15204	*FvCKX8*

* Previously reported CK biosynthesis [32], *ARR* types A, B, and C [33], and CK degradation [34] genes. The different background color in the header indicates the different type of genes (biosynthesis, signaling or degradation).

**Table 2 plants-12-03672-t002:** Exogenous application of cytokinins on commercial strawberries. Evaluation period, dosage, and frequency of application.

Fall–Winter Experiment (December 2020–February 2021)			Spring–Summer Experiment (April–July 2021)		
Treatment	Concentration ppm	Frequency of Application	Treatment	Concentration ppm	Frequency of Application
1	Control		1	Control	
2	50	1	2	50	1
3	50	2	3	50	3
4	50	3	4	100	1
5	50	4			
6	50	2			
7	50	1			

**Table 3 plants-12-03672-t003:** Homologous *F. vesca* and *F. × ananassa* genes.

F. vesca			F. × ananassa Blast in GDR			
Gene	ID	CDS	ID	Length pb	Identity	%
*CKX2*	FvH4_3g03260	1596	FxaC_9g52120.t1 FxaC_11g00140.t1	1578 1584	1576/1596 1548/1602	98.75 96.63
*CKX3*	FvH4_6g24620	1638	FxaC_21g36910.t1 FxaC_23g15950.t1	1638 1626	1634/1638 1609/1638	99.76 98.23
*CXK7*	FvH4_3g04610	1482	FxaC_9g50630.t1 FxaC_12g45970.t1 FxaC_10g02600.t1	1482 1809 1543	1472/1482 1424/1482 1420/1482	99.33 96.09 95.82
*IPT2*	FvH4_3g29650	1446	FxaC_12g15640.t1 FxaC_11g30460.t1	1857 1813	1425/1449 1420/1446	98.34 98.2

## Data Availability

Individual data regarding measurements of fruit dimensions and weight, and flower number obtained during the CK application experiments is available on request.

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
