# Peer review of "The Role of Cytokinins during the Development of Strawberry Flowers and Receptacles"

_plants, 2023, doi:10.3390/plants12213672_

Round 1
Reviewer 1 Report
This manuscript describes a series of experiments that include both database mining and empirical studies, to assess the importance of cytokinins in fruit development in strawberries and identify genes that it may be useful to manipulate (knock-out) for improving fruit production. Specifically:
1. The Fragaria homologs of gene important in cytokinin signaling in Arabidopsis were identified and analyzed at the amino acid sequence level and their expression patterns assessed using databased information.
2. Cytokinin content was assessed in fruit through its development.
3. The effects of exogenous cytokinin sprays were assessed on field grown plants.
4. Expression patterns of selected CKX genes that may be good targets for manipulation was studied using in situ hybridization.
The overall finding is that it does appear that cytokinin is important to fruit production in strawberries in several ways and manipulation of cytokinin pathways may have potential in improving production of this economically important crop. Hence these results have significance.
I would also like to note that this is one of the most carefully prepared manuscripts that I have reviewed in recent years – thank you authors!
There are just a few typographical errors that should be addressed and I have a couple of minor additional points.
1. Line 21 typo “acquenes” should be “achenes”
2. Line 24 typo “determing” should be “determining”
3. Line 319 “anacrostily” I have been working in plant reproduction for almost 40 years and I have never come across that term. Using Google, I eventually came across a reference (also on Fragaria) that used the term and referenced Weberling 1992, but it is a book and I could not find any details as to what the term means. It would be helpful if the authors could define the term for readers.
4. Line 470 “march should be “March”
5. Is yield data available? Presumably if you have fruit number (or do the authors just have flower number) and fruit weight it should be. It would be helpful to know if yield is changing or whether increased fruit size (an important trait commercially) is always offset by a reduction in fruit number (and vice versa).
6. The abbreviation DAA is used both as “days after anthesis” and “days after application” in different places in the manuscript. While it is possible to figure out when each is applicable, it is a little confusing when the reader first encounters the switch. It would be helpful to readers to use different abbreviations.....
Author Response
Dear reviewer, we appreciate the time you dedicated to reviewing this article and we also appreciate the comments, questions and recommendations you made to this work. Below we respond to each of your comments.
- Line 21 typo “acquenes” should be “achenes”
answer: Thank you, we have corrected this mistake in line 21.
- Line 24 typo “determing” should be “determining”
answer: It has been corrected in the text.
- Line 319 “anacrostily” I have been working in plant reproduction for almost 40 years and I have never come across that term. Using Google, I eventually came across a reference (also on Fragaria) that used the term and referenced Weberling 1992, but it is a book and I could not find any details as to what the term means. It would be helpful if the authors could define the term for readers.
answer: Thank you for this observation, to simplify the text we have removed the term “anacrostily”, which indeed did not add too much information.
- Line 470 “march should be “March”
answer: Thank you. It has been corrected in the text.
- Is yield data available? Presumably if you have fruit number (or do the authors just have flower number) and fruit weight it should be. It would be helpful to know if yield is changing or whether increased fruit size (an important trait commercially) is always offset by a reduction in fruit number (and vice versa).
answer: Thank you for the suggestion. We have included two graphs depicting the total number of fruits and the total weight obtained in the winter and summer experiments as supplementary data. We also have included some lines about this in the results section.
- The abbreviation DAA is used both as “days after anthesis” and “days after application” in different places in the manuscript. While it is possible to figure out when each is applicable, it is a little confusing when the reader first encounters the switch. It would be helpful to readers to use different abbreviations.....
answer: Thank you for this comment. To avoid confusion, in the exogenous application experiments, the abbreviations DAA were changed to DAFA (days after the first application). The change was made to the entire document.

Reviewer 2 Report
The manuscript is well-organized and written with sufficient data. There are only two minor issues.
1. In the abstract, it's advisable to include the main findings rather than just mentioning what needs to be done.
2. On line 267, the subtitle "2.3.1 Spring-summer experiment" isn't necessary, as the "fall-winter experiment" wasn't separated by a subtitle.
Author Response
Dear reviewer. Thank for your time, your kind comments, and the questions and recommendations. Below we respond to each of your comments.
- In the abstract, it's advisable to include the main findings rather than just mentioning what needs to be done.
answer: Thank you for this observation, we have modified the abstract to briefly describe results.
- comment/doubt/recommendation. On line 267, the subtitle "2.3.1 Spring-summer experiment" isn't necessary, as the "fall-winter experiment" wasn't separated by a subtitle.
answer: Thank you for this comment. We have removed the subtitle in the text.

Reviewer 3 Report
Dear authors,
Hope you are doing well.
I read your manuscript entitled "The role of cytokinins during the development of strawberry 2 flowers and receptacles". I think that the topic of study is highly relevant, however, I believe that some parts should be explained for a better overall understanding. Next, I indicate my concerns:
1. Introduction. This section seems too long. In my opinion, I think this should be reduced to 4-5 paragraphs. Furthermore, detailed information on the metabolism or signaling of cytokinins should be included in the discussion.
2. I suggest reviewing the information obtained in previous research that can complement the discussion of its results:
https://www.sciencedirect.com/science/article/abs/pii/S0889157521004622
https://www.sciencedirect.com/science/article/pii/S0981942823000359
3. The materials and methods section can be improved:
- Explain why you used 8 treatments in fall-winter and only 4 in spring-summer. Also, why did they use 100 ppm only in spring-summer or why some treatments were applied only 2 times, others 3, others 4,... In my opinion it is difficult to draw conclusions from the results when such different treatments are compared.
- Cytokinin content. You should explain better if to measure the concentration, they separated the achenes from the receptacle.
- The methodology of the phylogenetic analyses is not complete: the method used is lacking (NJ, MP,...), bootstrap values (also indicate in the phylogenetic trees).
4. It is not clear whether the signalling genes were reported by you or by other authors. Please review lines 153-158 and Table 1.
Best regards,
Author Response
Dear reviewer. We appreciate your time, coomments, questions and recommendations. Below we respond to each of your comments.
- comment/doubt/recommendation. This section seems too long. In my opinion, I think this should be reduced to 4-5 paragraphs. Furthermore, detailed information on the metabolism or signaling of cytokinins should be included in the discussion.
answer: Dear reviewer, thank you for your comment. We agree that the introduction is long for readers that are familiar with the topics presented. At the same time, for other readers that only familiar with one of the topics or that work in different fields, we considered that it was important to provide enough information for them to understand the results presented. This is also why the information about the cytokinin pathway was included in this section directly, so that the next section about gene families in the results would be clearer for them. Therefore, we removed some phrases from the introduction with the aim to make it more concise but left most information. We appreciate your suggestion and hope you can also see this other side.
- comment/doubt/recommendation. I suggest reviewing the information obtained in previous research that can complement the discussion of its results:
answer: Dear reviewer. Thank you for pointing out to these reports, they are very interesting. Please find our comments below:
- Analysis of multiple-phytohormones during fruit development in strawberry by using miniaturized dispersive solid-phase extraction based on ionic liquid-functionalized carbon fibers by Yang et al. The authors of this article carried out the study on Akihime variety and the strawberries were grown under controlled conditions. Yang et al. 2022 observed that cytokinin concentrations increased as fruit development progressed until maturity; in particular, they measured the zeatin content, but the text does not clarify whether it is cis or trans-zeatin. We did not quantify cis-zeatin, only trans-zeatin, and this could be a difference. This is very interesting, because there is little known about the biological role of cis-zeatin, in comparison to trans-zeatin. Moreover, though there is some overlap between the samples used, we also include stages that were not included in this study. Our determinations started with flower buds and flowers at anthesis, and continue with developing receptacles, and they focus on the latter only, and with more stages after anthesis. Finally, we do not know how much this could affect, but the authors used another variety and the growth conditions (protected), temperature, fertilization and management of the plants were different. The extraction method they use is very interesting and more sensitive. It would be interesting to consider it in further analyses.
- Comprehensive profiling of endogenous phytohormones and expression analysis of 1-aminocyclopropane-1-carboxylic acid synthase gene family during fruit development and ripening in octoploid strawberry (Fragaria× ananassa) by Upadhyay et al. They use the Monterrey variety, grown in the field (like us). This study measured 28 different hormones, including different cytokinin metabolic forms. They observed that isopentenyladenine content increased during ripening, but that most of the other forms decreased. This was the case for trans-zeatin (tZ), the form that we determined, and we also saw a decrease in content for this cytokinin form. They found that tZ content was very low at different stages of fruit development <100 pmol/g DW, regardless of the stage of development and the cytokinin evaluated. In our study, the lowest amount of transzeatin obtained was 160.5 pmol/g DW, while the highest was 5400 pmol/g DW in flower buds. However, flower buds are not included in the Upadhyay study. In summary, our results coincide regarding the diminishing content in developing fruits, and we also included different, earlier samples (flower buds and flowers in anthesis) where a high concentration of trans-zeatin was observed.
- We also have included some lines about these papers that you kindly indicate, in the discussion.
- comment/doubt/recommendation. Explain why you used 8 treatments in fall-winter and only 4 in spring-summer. Also, why did they use 100 ppm only in spring-summer or why some treatments were applied only 2 times, others 3, others 4,... In my opinion it is difficult to draw conclusions from the results when such different treatments are compared..
answer: Thank you for this observation. Seven treatments were used in winter and those that presented the highest values for fruit size and weight, and total number of fruits harvested were selected. For example, T4 with 3 applications of 50 ppm BA presented the highest values of length, width and fruit weight at 65 DAFA in the winter, so it was repeated in spring. In the case of T2 (one application of 50 ppm BA), it was the one that produced the highest number of fruits in total. We have included a new supplementary figure, Fig. S3 with this data that was not shown before, following the advice of reviewer 1. One application with 100 ppm BA was evaluated in both winter (T7) and spring (T4) as a comparison.
- comment/doubt/recommendation. Cytokinin content. You should explain better if to measure the concentration, they separated the achenes from the receptacle.
answer: Thank you for this observation. For this work, the amount of cytokinins was measured in flower buds and flowers in anthesis (where achenes are not yet formed, they are still ovaries or ovaries in development at these stages). In the case of developing fruits, the CK content of the receptacle together with achenes was measured, that is, the hormone was not measured in each tissue separately. We have included a brief note in the text to clarify better that achenes were present in the evaluated receptacle samples. This observation about the separation of achenes and receptacles is very valuable and will be taken into account in future work.
- comment/doubt/recommendation. The methodology of the phylogenetic analyses is not complete: the method used is lacking (NJ, MP,...), bootstrap values (also indicate in the phylogenetic trees).
answer: Thank you, we have included this information. For the phylogenetic trees, the MEGA 11 program was used with the Maximum Likelihood (MLE) method with 100 bootstraps. Figure 1 was modified to include the bootstrap values.
- comment/doubt/recommendation. It is not clear whether the signalling genes were reported by you or by other authors. Please review lines 153-158 and Table 1.
answer: Thank you for pointing this out. Table 1 mentions that the homologous genes of the biosynthesis, degradation and signaling pathways (RR type A, B and C) have already been described by other authors, although homologous genes of the HK and HP family have not yet been described. The homologous genes of the last two families are part of our contribution. We also have also included this information in the text to clarify this further.

Round 2
Reviewer 2 Report
The revised manuscript got further improvement.
Reviewer 3 Report
Dear authors,
Thank you very much for your responses. I understand your arguments regarding the introduction section. Therefore, I also consider that it may be important to detail this information. Regarding the rest of the changes made to the first version, I think the manuscript has improved a lot.
Best regards,